# Identification and New Indication of Melanin-Concentrating Hormone Receptor 1 (MCHR1) Antagonist Derived from Machine Learning and Transcriptome-Based Drug Repositioning Approaches

**DOI:** 10.3390/ijms23073807

**Published:** 2022-03-30

**Authors:** Gyutae Lim, Ka Young You, Jeong Hyun Lee, Moon Kook Jeon, Byung Ho Lee, Jae Yong Ryu, Kwang-Seok Oh

**Affiliations:** 1Data Convergence Drug Research Center, Korea Research Institute of Chemical Technology, 141 Gajeong-ro, Yuseong-gu, Daejeon 34114, Korea; gyutae@krict.re.kr (G.L.); uuuky@krict.re.kr (K.Y.Y.); jhlee@krict.re.kr (J.H.L.); moteta@krict.re.kr (M.K.J.); bhlee@krict.re.kr (B.H.L.); 2Graduate School of New Drug Discovery and Development, Chungnam National University, Daejeon 34134, Korea; 3Department of Biotechnology, Duksung Women’s University, 33 Samyang-ro 144-gil, Dobong-gu, Seoul 01369, Korea; 4Department of Medicinal and Pharmaceutical Chemistry, University of Science and Technology, 217 Gajeong-ro, Yuseong-gu, Daejeon 34113, Korea

**Keywords:** melanin-concentrating hormone, MCHR1 antagonists, obesity, cardiotoxicity, hERG, machine learning model, cholesterol reduction

## Abstract

Melanin-concentrating hormone receptor 1 (MCHR1) has been a target for appetite suppressants, which are helpful in treating obesity. However, it is challenging to develop an MCHR1 antagonist because its binding site is similar to that of the human Ether-à-go-go-Related Gene (hERG) channel, whose inhibition may cause cardiotoxicity. Most drugs developed as MCHR1 antagonists have failed in clinical development due to cardiotoxicity caused by hERG inhibition. Machine learning-based prediction models can overcome these difficulties and provide new opportunities for drug discovery. In this study, we identified KRX-104130 with potent MCHR1 antagonistic activity and no cardiotoxicity through virtual screening using two MCHR1 binding affinity prediction models and an hERG-induced cardiotoxicity prediction model. In addition, we explored other possibilities for expanding the new indications for KRX-104130 using a transcriptome-based drug repositioning approach. KRX-104130 increased the expression of low-density lipoprotein receptor (LDLR), which induced cholesterol reduction in the gene expression analysis. This was confirmed by comparison with gene expression in a nonalcoholic steatohepatitis (NASH) patient group. In a NASH mouse model, the administration of KRX-104130 showed a protective effect by reducing hepatic lipid accumulation, liver injury, and histopathological changes, indicating a promising prospect for the therapeutic effect of NASH as a new indication for MCHR1 antagonists.

## 1. Introduction

Melanin-concentrating hormone (MCH) is a circulating neuropeptide that regulates feeding behavior and energy balance in mammals. It is known to be expressed throughout the brain, with its highest levels in the lateral hypothalamus and zona incerta [1,2]. The mRNA levels of MCH in the hypothalamus are upregulated in leptin-deficient mice and further increased by fasting [3,4]. Melanin-concentrating hormone deficiency is hypophagic and hypermetabolic, leading to weight loss and increased slenderness, whereas overexpression of MCH leads to obesity and insulin resistance [5]. Pharmacological validation of these studies suggests that melanin-concentrating hormone receptor 1 (MCHR1) antagonists that mediate MCH regulation could be used as appetite suppressants for obesity.

Previously, we developed novel MCHR1 antagonists such as 2-arylbenzimidazole derivatives [6], 4-arylphtalazin-1(2h)-one derivatives [7], and pyrrolo(3,4-b)pyridine-7(6H)-one derivatives [8]. These compounds strongly antagonize the MCHR1 receptor with highly potent and favorable pharmacokinetics. However, the structure of the drug developed as an MCHR1 antagonist is very similar to that of the human Ether-à-go-go-Related Gene (hERG) blocker [9], which is likely to cause cardiotoxicity problems, and their preliminary efficacy data from a proof-of-concept program were unsatisfactory. Therefore, subsequent drug development of the MCHR1 antagonist did not proceed.

Recently, computational methods have been actively applied as efficient tools for drug screening and expanding indications. In particular, machine learning-based methods such as random forest (RF), support vector machine (SVM), deep neural network (DNN), and graph convolution network (GCN) can rapidly predict drug activity, even in large quantities of compounds [10,11,12], and cardiotoxicity [13,14]. A transcriptome-based method was used to select targets of the screened molecules. As the gene expression by the administered drug differs depending on the experimental group, it is possible to find genes that are significantly expressed through a differentially expressed gene (DEG) analysis in comparison to that of the control group [15,16,17]. Through Gene Ontology (GO) [18] and Kyoto Encyclopedia of Genes and Genomes (KEGG) [19] enrichment analyses, the biological functions and mechanisms of target proteins were identified. In addition, using datasets such as The Cancer Genome Atlas (TCGA) [20], Cancer Cell Line Encyclopedia (CCLE) [21], and the Library of Integrated Network-based Cellular Signatures (LINCS) [22] which have drug-disease gene expression profiles, it has become possible to identify targets for specific diseases, cell lines, and perturbations under more precise conditions [23,24,25,26].

In this study, we attempted to discover novel MCHR1 antagonists without cardiotoxicity using DNN-based machine learning models and to identify new indications by analyzing gene expression. The constructed models were hypothesized to successfully predict MCHR1 activity for 357,842 compounds in the Korea Chemical Bank database, and we further used the cardiotoxicity prediction model [13] to derive the final compound KRX-104130 that overcame cardiotoxicity. In addition, we aimed to analyze the effect of KRX-104130 on the low-density lipoprotein receptor (*LDLR*) gene [27,28], which affects cholesterol reduction, using a transcriptome-based drug repositioning approach [15] and to show presence of any protective effects in a mouse model of nonalcoholic steatohepatitis (NASH).

## 2. Results

### 2.1. Development of Prediction Model for MCHR1 Binding Activity

To discover MCHR1 antagonists efficiently, virtual screening was performed. As the crystal structure of MCHR1 was not available, ligand-based virtual screening was used instead of protein structure-based virtual screening. To build ligand-based machine learning models that predict MCHR1 binding activity (i.e., log-transformed IC_50_), we first prepared an in-house dataset containing compounds with MCHR1 binding activity from the Korea Chemical Bank (see Section 5). In total, 1065 compounds with information on MCHR1 binding activity were identified in the in-house dataset. For model development, the in-house dataset was split into three sub-datasets: training (80%), validation (10%), and test (10%) for model development (see Section 5.3 for details).

In this study, we used two types of molecular features (molecular descriptor- and molecular fingerprint-based features). Thus, two independent machine learning models based on each molecular feature were constructed: descriptor-based and fingerprint-based DNN models (Figure 1). We tested 63 sets of molecular features for each DNN to determine the optimal number of features (i.e., molecular descriptors and molecular fingerprints). The descriptor- and fingerprint-based DNN models showed the best performance when using 500 molecular descriptors and 800 molecular fingerprints, respectively (Appendix A). In addition, we performed hyperparameter optimizations to determine the best model with the highest mean squared error (MSE) on the validation dataset. The final descriptor- and fingerprint-based DNN models showed MSE values of 0.072 and 0.076 and R^2^ values of 0.660 and 0.640 for the on the test dataset, respectively (Appendix A).

### 2.2. Virtual Screening of MCHR1 Antagonists Using Ligand-Based Machine Learning Models

We used two machine learning models (molecular descriptor- and molecular fingerprint-based DNN models) for the virtual screening of MCHR1 antagonists. For this purpose, we used the chemical library of the Korea Chemical Bank, which contains 357,842 compounds. The molecular descriptor- and molecular fingerprint-based models were used to predict the MCHR1 binding activity (i.e., log-transformed IC_50_) for all compounds in the chemical library, respectively. Then, the top 300 compounds with high MCHR1 binding activity predicted by each prediction model were selected. Through virtual screening, 592 compounds were found after excluding duplicates. To reduce the number of candidate compounds, we considered only those compounds with at least a Tanimoto similarity ≥0.4, with the structures of MCHR1 antagonists present in the training data. As a result, 479 of 592 compounds were removed, leaving 113 compounds. MCHR1 antagonists are likely to exhibit hERG-induced cardiotoxicity because MCHR1 antagonists are structurally similar to hERG ligands. To avoid this problem, we predicted the hERG-induced cardiotoxicity of 113 compounds using DeepHIT [13], which is a computational framework for predicting hERG-induced cardiotoxicity (i.e., hERG blockers) based on three independent deep neural networks. As a result, only 19 compounds were predicted to be hERG non-blockers. Thus, these compounds were used further to validate MCHR1 binding activity.

### 2.3. Identification of Small-Molecule Antagonist of MCHR1

An in vitro time-resolved fluorescence (TRF)-based binding assay for MCHR1 was performed to identify the MCHR1 antagonist from the 19 predicted compounds selected through in silico screening. Compounds with low reproducibility, poor dose–response profiles, and non-specific binding activities were eliminated, and the 1-(6,7-Difluoro-3-methoxyquinoxalin-2-yl)-3-[1-(heteroarylmethyl)piperidin-4-yl] urea derivatives exhibited potent antagonistic activity. Among them, five compounds (KRX-104130, KRX-104137, KRX-104156, KRX-104161, and KRX-104165) showed the strongest potency; the inhibition rates (%) at 10 μM were 97.1%, 96.0%, 100.1%, 94.8%, and 94.3%, and the IC_50_ values were 0.02 μM, 0.01 μM, 0.05 μM, 0.01 μM, and 0.06 μM, respectively (Figure 2). In addition, a patch clamp assay was performed to measure the cardiotoxicity of the five compounds. The IC_50_ values for the hERG channel of the compounds were 12.98 μM, 0.42 μM, 0.92 μM, 0.18 μM, and >50 μM, respectively (Appendix A). Consequently, of the five compounds, only KRX-104130 and KRX-104165 overcame cardiotoxicity.

### 2.4. Plasma Pharmacokinetic (PK) Profile of KRX-104130

KRX-104130 and KRX-104165, which showed no cardiotoxicity, were administered to mice to confirm their in vivo action. For KRX-104130, after intravenous (I.V.) administration of 10 mg/kg to three mice, the clearance rate (CL) was 2.21 ± 0.11 L/kg/h, the area under the curve (AUC_t-0_) was 4.55 ± 0.22 μg∙h/mL, the half-life (T_1/2_) was 5.71 ± 1.03 h, and the steady-state volume of distribution (Vss) was 14.16 ± 0.72 L/kg. In addition, after the oral (P.O.) administration of 20 mg/kg of KRX-104130 to three mice, the peak concentration reached to 6.67 ± 2.31 h, the AUC_t-0_ was 1.72 ± 0.85 μg∙h/mL, the T_1/2_ was 10.22 h, the maximum concentration (C_max_) was 0.13 ± 0.05 μg/mL, and the bioavailability (F) was 20.3% (Table 1). Conversely, KRX-104165 was excluded from the candidate compound because the mouse died immediately after administration. Therefore, KRX-104130 was selected for further analyses.

### 2.5. mRNA Gene Expression Analysis

Transcriptome analysis was performed to identify new indications of KRX-104130. For each of the four cell lines, human skin malignant melanoma cells (A375), human lung carcinoma cells (A549), human breast carcinoma cells (MCF7), and human prostate adenocarcinoma cells (PC3), 10 μM of KRX-104130 was administered for 6 h in the control and treatment groups. As an independent experiment was performed three times, considering the error, the average value was used for comparison. The change in gene expression levels was calculated by dividing the gene expression value of the control group by the gene expression value of the treatment group. Fold change (FC) was expressed as a positive or negative value when the gene expression value of the treatment group was greater or lower than that of the control group, respectively.

We analyzed the DEGs in each cell line based on the calculated gene expression values (Figure 3). There were 137 upregulated genes and 43 downregulated genes in the A375 cells, and 216 upregulated genes and 87 downregulated genes in the A549 cells. In addition, 73 upregulated genes and 23 downregulated genes were identified in the MCF7 cells and 143 upregulated genes and 93 downregulated genes in the PC3 cells. For all cell lines, we identified eight common genes with increased expression (*CCNG2*, *DNAJB9*, *HERPUD1*, *HSD17B7*, *INSIG1*, *LDLR*, *MSMO1*, and *TIPARP*) and one with decreased expression (*CCNE2*).

### 2.6. GO and KEGG Pathway Enrichment Analyses

After excluding duplicates from the four cell lines, the total number of DEGs was 689 (460 upregulated and 229 downregulated) genes. We performed GO enrichment and KEGG pathway enrichment analyses using the Database for Annotation, Visualization, and Integrated Discovery (DAVID) to identify the mainly expressed functions and pathways. The DEGs used for the GO and KEGG enrichment analyses also satisfied |FC| ≥ 1.5 and *p* < 0.05. In the GO enrichment analysis, only the BP GO term was used to confirm the biological processes.

In the GO enrichment analysis of 416 upregulated genes having the valid BP GO terms, the top five BP GO terms were positive regulation of transcription from the RNA polymerase II promoter (10.2%), negative regulation of transcription from the RNA polymerase II promoter (9.6%), oxidation-reduction process (6.1%), positive regulation of transcription, DNA-templated synthesis (5.2%), and positive regulation of GTPase activity (5.0%). In contrast, the top five BP GO terms of the 213 downregulated genes with the valid BP GO terms were positive regulation of transcription from the RNA polymerase II promoter (11.9%), negative regulation of transcription from the RNA polymerase II promoter (9.3%), negative regulation of apoptotic process (8.4%), positive regulation of transcription, DNA-templated synthesis (6.6%), and negative regulation of cell proliferation (6.2%). There was no significant difference between the upregulated and downregulated genes in the BP GO terms. The expression of genes involved in transcriptional regulation was prominent (Figure 4).

In the KEGG enrichment pathway analysis, the top five pathways for the 212 upregulated genes were protein processing in the endoplasmic reticulum (4.4%), biosynthesis of antibiotics (4.4%), transcriptional misregulation in cancer (3.1%), insulin resistance (2.6%), and non-alcoholic fatty liver disease (NAFLD) (1.2%). Conversely, the top five pathways of the 109 downregulated genes were pathways in cancer (6.6%), phosphatidylinositol 3-kinase (PI3K)-protein kinase B (Akt) signaling pathway (6.2%), mitogen-activated protein kinase (MAPK) signaling pathway (4.8%), Hippo signaling pathway (4.0%), and proteoglycans in cancer (4.0%) (Figure 5).

In addition, GO and KEGG analyses were performed on the genes commonly expressed in all four cell lines. The GO terms for the eight upregulated genes were cholesterol biosynthetic processes (37.5%), cholesterol metabolic processes (25.0%), and palate development (25.0%). Although the GO term for *CCNE2*, a downregulated gene, could not be found, it controls the cell cycle in the late G1 and early S phases. In the KEGG pathway of the common expression genes, steroid biosynthesis (25.0%) and ovarian steroidogenesis (25.0%) were upregulated genes, but there was no pathway in the downregulated gene (Appendix A).

When we considered the GO terms and KEGG pathways from all cell lines and commonly expressed genes together, the property related to the new indication of an MCHR1 antagonist was cholesterol regulation. Pathway analysis results inferred that the related disease was NAFLD.

### 2.7. Comparison of KRX-104130-Induced Gene Expression in NASH Patients

KRX-104130 was confirmed to affect genes related to insulin resistance and cholesterol metabolism through the GO enrichment analysis and was closely related to specific diseases such as NAFLD in the KEGG enrichment analysis. NAFLD affects approximately a quarter of the adult population worldwide, and 10% of NAFLD cases progress to NASH and, in the worst case, liver cancer. In addition, clinical studies on the use of MCHR1 antagonists in NASH treatment are also in progress [29].

We compared the NASH patient group (GSE89632, https://www.ncbi.nlm.nih.gov/geo/query/acc.cgi?acc=GSE89632, accessed on 1 September 2021) from the Gene Expression Omnibus (GEO) with nine genes that had a common effect on the four cell lines to identify the gene expression patterns in the NASH patient group. The difference in gene expression in the NASH patient group was confirmed by calculating the FC in the same manner as in KRX-104130. As shown in Figure 6, *LDLR* showed the most significant change (FC = −1.65) among the nine commonly expressed genes in the NASH patient group, and the other genes showed no significant change in expression level. In contrast, in KRX-104130-induced *LDLR*, the FC increased to 2.79. These results indicate that it is possible to increase the expression of *LDLR* that is reduced by KRX-104130.

### 2.8. The Effects of KRX-104130 on the mRNA and Protein Expression Levels of LDLR

To validate that KRX-104130 increased the *LDLR* gene expression through the DEG analysis, *LDLR* mRNA expression was investigated in KRX-104130–treated HepG2 cells. After 24 h of KRX-104130 exposure, the HepG2 cells exhibited *LDLR* upregulation in a concentration-dependent manner of KRX-104130 (Figure 7A). Consistent with this result, *LDLR* protein levels increased in a concentration-dependent manner of KRX-104130 (Figure 7B).

### 2.9. Effect of KRX-104130 in a NASH Animal Model

We examined whether the pathological changes in the liver were attenuated by KRX-104130 in a methionine–choline-deficient (MCD) diet induced NASH mouse model because a recent study showed that upregulation of hepatic LDLR is associated with amelioration of MCD-induced NASH features [30]. The MCD diet significantly increased the liver triglyceride (TG) levels compared with that in the methionine–choline-sufficient (MCS) control group (41.71 ± 1.82 vs. 29.82 ± 2.20, *p* < 0.05). The levels of liver damage markers such as aspartate aminotransferase (AST) and alanine aminotransferase (ALT) were also significantly increased in the MCD diet group compared with that in the control group (274.50 ± 38.8 vs. 49.2 ± 2.76; 267.0 ± 25.52 vs. 24.0 ± 1.64, *p* < 0.05, respectively). The KRX-104130-treated group showed a significant decrease in liver TG levels (29.73 ± 1.54, *p* < 0.05) and in serum concentrations of AST (170.5 ± 27.27, *p* < 0.05) and ALT (163.67 ± 25.85, *p* < 0.05) compared with those of the vehicle group (Figure 8A–C). In addition, the MCD diet-induced histopathological changes, such as steatosis and fibrosis in liver tissues, were ameliorated by treatment with KRX-104130 (Figure 8D).

## 3. Discussion

Melanin-concentrating hormone regulates feeding behavior, and MCHR1 is an essential protein that mediates this behavior. Therefore, MCHR1 antagonists that target MCHR1 have been developed as representative appetite suppressants. However, given that the structure of MCHR1 antagonists is very similar to that of the hERG ligand, it cannot overcome cardiotoxicity, making it challenging to develop. Unlike other discontinued drugs, we aimed to find novel MCHR1 antagonists without cardiotoxicity and explore new possibilities.

An important condition for finding an MCHR1 antagonist is sufficient activity to overcome the cardiotoxicity. Given that it was impossible to experimentally confirm all binding activities in the database of 357,842 compounds, we applied a computational method. We built two deep learning-based prediction models to screen the MCHR1 antagonists using MCHR1 binding activity data. Thus, it was possible to learn the structural characteristics and activity of the compound, and satisfactory prediction data were obtained using the optimized model. The constructed prediction models predicted MCHR1 binding affinity and cardiotoxicity using DeepHIT [13]. Therefore, 19 candidate compounds were selected through virtual screening and were experimentally confirmed to be effective. Among the predicted compounds, two (KRX-104130 and KRX-104165) overcame cardiotoxicity and, finally, preliminary efficacy data was derived. However, in a pharmacokinetic study, KRX-104165 caused unexpected death due to unexpected toxicities when administered to mice. These unexpected deaths could be explained by toxicity of major metabolite, off-target activity due to the difference in heteroaryl group. As shown in Figure 2 and Appendix A, the indole moiety of KRX-104165 and KRX-104130 make to overcome the cardiotoxicity of 1-(6,7-Difluoro-3-methoxyquinoxalin-2-yl)-3-[1-(heteroarylmethyl)piperidin-4-yl]urea scaffold while maintaining the MCHR activity. However, in the machine learning-based predictive analysis of metabolic stability [31], KRX-104165 with 3-acetylindole moiety was predicted to be relatively unstable metabolic stability than KRX-104130 with an indole moiety alone. Therefore, it is likely that the major metabolite isolated from KRX-104165 induces cardiotoxicity. However, due to the limitations of the in silico predictive analysis, further studies are needed. On the other hand, KRX-104130 showed good bioavailability for oral administration without specific abnormalities. We explored new possibilities for KRX-104130, which, unlike other discontinued drugs, was safe for toxicity issues and exhibited a good pharmacokinetic profile.

In this case, drug repositioning can be effectively utilized. We attempted to identify new indications using gene expression analysis. After KRX-104130 treatment of the four cancer cell lines, changes in gene expression were confirmed through mRNA experiments. The GO and KEGG pathway enrichment analyses using DEGs confirmed that KRX-104130 significantly affected cholesterol biosynthesis and metabolic processes, particularly insulin resistance and steatohepatitis. However, given that there was a limitation in identifying direct targets affecting insulin resistance and steatohepatitis of KRX-104130 in the transcriptome analysis, we compared with the gene expression of NASH patients.

Nine genes commonly affected by KRX-104130 showed no significant changes in expression in the NASH patient group. However, the *LDLR* was significantly reduced compared with that in the normal group. In contrast, KRX-104130 treatment significantly increased *LDLR* expression. Based on the relationship between *LDLR* and NASH, we identified that NASH symptoms differ depending on the presence or absence of *LDLR* expression [30,32,33,34,35]. In addition, the clinical results of applying MCHR1 antagonists to NASH were confirmed, and these results provided evidence that KRX-104130 could be used for NASH.

Based on these prediction results, the genes and proteins expressed by KRX-104130 were experimentally confirmed. In addition, the therapeutic potential of KRX-104130 was examined in a NASH animal model. If KRX-104130 lowered the cholesterol levels by increasing the *LDLR* levels, it was expected that the proportion of fat in the mouse liver cells would decrease. In the comparison of the NASH mouse model, lipid droplets size and fibrosis decreased in the liver treated with KRX-104130. In the case of a liver with steatohepatitis, the color of the liver changed to yellow as the size of fat cells increased. As the size of the fat cells decreased, the color changed to red.

Fat accumulation in the liver leads to inflammation which damages the liver cells. Therefore, it is essential to prevent the accumulation of fat to eliminate the cause of inflammation. LDLR is a protein involved in cholesterol metabolism and helps relieve cholesterol by trapping LDL. In addition, oxLDL, known as a biomarker of NASH, is emerging as a new risk factor for liver inflammation [36]. As oxLDL is produced by the oxidation of LDL, an increase in LDLR, which can lower LDL, can be a clear benefit in NASH treatment.

These results indicated that KRX-104130 effectively acts on NASH and could alleviate symptoms partly by increasing *LDLR*. Therefore, our approach was able to rapidly screen compounds in the drug database at the initial stage and effectively predict the point of action of drugs through genomic analysis. In addition, it is of great significance because it provides clues for a new indication that the MCHR1 antagonist could be expected to have a therapeutic effect on NASH.

## 4. Conclusions

We used two DNN-based MCHR1 binding affinity prediction models and a drug repositioning approach using genomic analysis to identify new indications of MCHR1 antagonists. Although it was difficult for the compound predicted to have good activity in the candidate-screening step to act as an appetite suppressant in vivo, nine commonly expressed genes were identified by DEG analysis. Through GO/KEGG analysis, KRX-104130 was found to be insulin resistant, and it was found to influence NASH by being involved in cholesterol regulation. In addition, comparison of the gene expression of the NASH patient group showed that the characteristically expressed gene was *LDLR* and that the decreased gene expression was increased by KRX-104130. Moreover, hepatic lipid accumulation and liver injury were significantly reduced by KRX-104130 treatment in a NASH mouse model. These results indicated that KRX-104130 can be used in effective treatment for NASH. Machine learning and transcriptome-based methods are expected to be useful in drug repositioning research to identify new indications. In summary, KRX-104130, an MCHR1 antagonist that overcame cardiotoxicity, could regulate liver levels by inducing an increase in *LDLR*, which is involved in cholesterol regulation and provides helpful clues for NASH treatment.

## 5. Materials and Methods

### 5.1. Data Preparation

Compounds with MCHR1 binding activity were obtained from the Korea Chemical Bank. In total, 1065 compounds with MCHR1 binding activity (IC_50_) were applicable for model training. When the IC_50_ value was >10, the maximum IC_50_ value was fixed at 10. The MCHR1 binding activities of the 1065 compounds were log-transformed as follows: log(IC_50_+1). The structures of the compounds were presented in the simplified molecular-input line-entry system (SMILES) format, which is a specification in the form of a line notation to describe the structure of the chemical compounds. All chemical structures were standardized using the Python package RDKit (http://www.rdkit.org, accessed on 1 July 2021) and MolVS (https://github.com/mcs07/MolVS, accessed on 1 July 2021). The standardization process included removing small fragments, ionizing, and calculating the stereochemistry. The dataset was then split into three subdatasets: training (80%), validation (10%), and test (10%). The training dataset was used to train the models, the validation dataset was used to optimize the hyperparameters, and the test dataset was used to evaluate the prediction performance of the final models.

### 5.2. Preparation of Molecular Features

For preparation of molecular features, we used molecular descriptors and molecular fingerprints. Molecular descriptors represent structural and physicochemical features of compounds, and molecular fingerprints represent molecular structures in the form of binary bits that indicate the presence or absence of specific substructures. Molecular descriptors for each compound were calculated using the Python package Mordred [37]. Molecular descriptors that could be calculated for all compounds were not considered. Thus, 1440 molecular descriptors were used for the model development. The values of each molecular descriptor were normalized to the range of 0–1. Molecular fingerprints for each compound were calculated using the Python package PyBioMed [38]. For this purpose, extended-connectivity fingerprints with a maximum diameter of 2 (ECFP2) [39], and PubChem fingerprints were used. In total, 1024 extended-connectivity fingerprints (ECFPs) and 881 PubChem fingerprints were generated for each compound and used for model development.

### 5.3. Model Development

We used two independent models: a descriptor-based DNN and fingerprint-based DNN. A DNN is a multilayered neural network consisting of an input layer, a fully connected hidden layer(s), and an output layer [40]. The two neural networks used in this study were regression models that predicted MHCR1 binding activity for a given compound.

A Bayesian optimization algorithm was used to determine the best hyperparameter setting [41]. Bayesian optimization was implemented using the Python package skopt (https://scikit-optimize.github.io, accessed on 1 July 2021). The following hyperparameters were considered for optimization (see Appendix A for details): dropout rate, optimizer, learning rate, L2 regularization rate, batch size, number of fully connected hidden layers, and number of hidden nodes in each hidden layer. In this study, 63 sets of molecular features were tested for each DNN to determine the optimal number of molecular features. The descriptor-based DNN exhibited the best performance when using 500 molecular descriptors, and the fingerprint-based DNN exhibited the best performance when using 800 molecular fingerprints (Appendix A). In addition, we tested 100 sets of hyperparameters for both the descriptor-based and fingerprint-based DNNs. Among the 100 sets of hyperparameters, we selected the best hyperparameter setting for each DNN, showing the minimum MSE for the validation dataset (Appendix A).

The MSE was used as the loss function for model training. The optimized descriptor- and fingerprint-based DNN models were trained for up to 50 epochs (Appendix A). To prevent overfitting, dropout and L2 regularization methods were used. Model training was implemented using the Python package TensorFlow (version 1.13.1) [42]. A workstation with an Intel Xeon Gold 6150 (18 core processors, 2.7 GHz) CPU and an NVIDIA Tesla V100 graphics-processing unit was used for the model training.

### 5.4. MCHR1 Binding Assay 

The receptor-binding affinities of the generated 19 compounds were determined by time-resolved fluorescence (TRF)-based receptor-binding assays using europium-labeled MCH (Eu-MCH) in 96-well AcroWell™ plates. The MCH labeled with europium at the N-1 position was supplied by the Wallac Labeling Service (PerkinElmer, Inc., Waltham, MA, USA). The human recombinant MCH-1 receptor membrane preparation (MCH-1/SLC1 membrane) was obtained from Euroscreen S.A. (PerkinElmer). The binding assay buffer contained 25 mM 4-(2-hydroxyethyl)-1-piperazineethanesulfonic acid (HEPES; pH 7.4), 5 mM MgCl_2_, 1 mM CaCl_2_, and 0.5% bovine serum albumin. Total binding was determined from the binding reaction in the presence of 2 nM Eu-MCH and 5 μg/unit MCH-1 receptor membrane for 90 min. Nonspecific Eu–MCH binding was defined in the presence of 0.5 µM unlabeled MCH. After incubation for 90 min, the plate was washed three times with 300 µL of ice-cold 25 mM HEPES buffer (pH 7.4) in an automatic vacuum filtration system. The Eu was dissociated from the bound ligand by the addition of 150 µL of DELFIA enhancement solution (Perkin Elmer) and incubated for 10 min with shaking. Fluorescence signals from dissociated Eu were measured using a multilabel reader (Envision, PerkinElmer) with an excitation wavelength (340 nm) and an emission wavelength (615 nm).

### 5.5. Patch Clamp Assay 

Patch clamp assays were performed on HEK293 cells stably expressing the hERG channel (HEK293-hERG cells; Genionics, Switzerland). When the cells were more than 80% confluent, they were freshly dislodged from flasks with 0.05% trypsin–EDTA for 1–2 min, spun down twice at 700× *g*, suspended into the extracellular solution (in mmol/L at 1.8 × 10^6^ per mL: 137 NaCl, 4 KCl, 1 MgCl_2_, 1.8 CaCl_2_, 10 HEPES, and 10 glucose, pH 7.4, adjusted with NaOH), and dispensed into a patch clamp plate. The cell density was adjusted to 2–3 × 10^5^ cells/mL. After dispensing, the cells were applied to the AutoPatch system (PatchXpress 7000A, Molecular Devices, Sunnyvale, CA, USA). The hERG currents were evoked by two voltage pulses at 3 s interval. The voltage pulse consisted of a 100-ms step to −30 mV, a conditioning prepulse (2 s duration, +25 mV or +45 mV), followed by a test pulse (2 s duration, −30 mV) from a holding potential of −70 mV. After a 3 s interval at −70 mV, a second pulse protocol was applied, consisting of a 100-ms step to −30 mV, a prepulse (2 s duration, +45 mV), followed by a test pulse (2 s duration, −30 mV). Leak currents were linearly subtracted, extrapolating the current elicited by a 100-ms step to −80 mV from a holding potential of −70 mV. The experiment was initiated with a buffer control in the absence of the drug to calculate the IC_50_. The test compound was applied at low to high concentrations, and the tail current was monitored continuously. The compound effects were tested using six concentrations in three-fold dilutions from 1 to 300 µM.

### 5.6. Cell Culture and Sample Preparation for Gene Expression Analysis

Four cell lines were used for the cell culture: A375, A549, MCF7, and PC3, purchased from the American Type Culture Collection (ATCC, Rockville, MD, USA). The A375 and A549 cells were cultured in RPMI-1640 medium supplemented with 10% fetal bovine serum (FBS) and 1% penicillin-streptomycin-glutamine. The MCF7 cells were cultured in Dulbecco’s modified Eagle’s medium (DMEM) with 10% FBS and 1% penicillin-streptomycin-glutamine. The PC3 cells were cultured in RPMI with 10% FBS, 1% penicillin-streptomycin-glutamine, 1 mM sodium pyruvate, and 10 mM HEPES. KRX-104130 was synthesized at the Research Center for Medicinal Chemistry, Korea Research Institute of Chemical Technology (KRICT, Daejeon, Korea).

After releasing the stock, the cells were used in the experiments and cultured for two weeks to stabilize. KRX-104130 was dissolved in dimethylsulfoxide (DMSO) to form a 10 mM stock solution and stored in a deep freezer (−80 °C). Six sheets were seeded in each of the four cell lines, three sheets of DMSO and KRX-104130 (DMSO × 3 and KRX-104130 × 3), and they were stored in an incubator for 24 h. Cells were plated at 1.1375 × 10^6^ cells for A375, 1.3 × 10^6^ cells for A549, 8.45 × 10^5^ cells for MCF7, and 1.496 × 10^6^ cells for PC3 under the culture conditions. After 24 h, DMSO and KRX-104130 were diluted 1000-fold in the medium, and the cells were treated with a final concentration of 0.1% DMSO and 10 μM KRX-104130. After 6 h, the medium was removed, and the cells were collected by treatment with 1 mL of TRIzol. After destroying the cells by pipetting, they were stored in a deep freezer (−80 °C).

### 5.7. mRNA Expression Data Analysis

The total RNA samples were assessed using the Clariom™ S Assay Human Platform. cDNA was synthesized using the GeneChip WT (Whole Transcript) Amplification Kit as described by the manufacturer. The sense cDNA was then fragmented and biotin-labeled with terminal deoxynucleotidyl transferase (TdT) using the GeneChip WT Terminal labeling kit. Approximately 5.5 μg of labeled DNA target was hybridized to the Affymetrix GeneChip Array at 45 °C for 16 h. The hybridized arrays were washed and stained on a GeneChip Fluidics Station 450 and scanned using a GCS3000 scanner. Probe cell intensity data computation and CEL file generation were performed using Affymetrix^®^ GeneChip Command Console^®^ Software version 6.0+ (AGCC, www.thermofisher.com/kr/ko/home/life-science/microarray-analysis/microarray-analysis-instruments-software-services/microarray-analysis-software/affymetrix-genechip-command-console-software.html, accessed on 1 September 2021). The CEL file is a binary format that contains the intensity calculated result for the pixel values of the DAT file and is used to extract normalized intensity to be used for actual analysis in Affymetrix^®^ Power Tools version 2.11.4 (APT, www.thermofisher.com/kr/en/home/life-science/microarray-analysis/microarray-analysis-partners-programs/affymetrix-developers-network/affymetrix-power-tools.html, accessed on 1 September 2021). The normalized intensity was extracted from the data by performing background subtraction, normalization, and summarizing probe sets from the CEL file using the signal space transformation-robust multichip analysis (SST-RMA) algorithm of the Affymetrix^®^ Power Tools.

The statistical significance of the expression data was determined using the LPE test and FC, in which the null hypothesis was that no difference existed among groups. The FC and *p*-value cutoff were 1.5 and 0.05, respectively. The DAVID functional annotation tools [43,44] were used to calculate gene enrichment, pathways, and functional annotation analysis for a significant probe list.

### 5.8. Real-Time Quantitative Transcription PCR (RT-qPCR) Analysis for LDLR mRNA Expression Level

The mRNA expression levels of *LDLR* in the HepG2 cells were determined by real-time quantitative PCR (RT-qPCR) at the Cosmo Genetech (Sungsoo, Seoul, Korea). The total RNA was extracted from the cultured cells using an RNeasy mini kit (Qiagen, Valencia, CA, USA), and gene expression was analyzed by RT-qPCR using a Bio-Rad CFX96 system (Hercules, CA, USA). The mRNA expression levels were investigated three times in each sample treated with 0 μM, 3 μM, 10 μM, and 30 μM KRX-104130 for 24 h, and each result was obtained from three independent experiments, for 12 samples. For *LDLR*, the forward primer was 5′-CTC CCG CCA AGA TCA AGA AA-3′, and the reverse primer was 5′-GAG ATC TAG GGT GAT GCC ATT G-3′. The housekeeping gene, glyceraldehyde 3-phosphate dehydrogenase (GAPDH), was used as an internal control. The forward primer was 5′-TTG CCA TCA ATG ACC CCT TCA-3′, and the reverse primer was 5′-CGC CCC ACT TGA TTT TGG A-3′. The PCR conditions for all genes were 40 cycles of pre-denaturation at 95 °C for 3 min, amplification at 62 °C for 30 s, and 95 °C for 10 s. The qPCR was performed using CFX manager (Version 3.1).

### 5.9. Western Blot Assay for LDLR Protein Expression Level

The protein expression levels of *LDLR* in the HepG2 cells were determined by Western blot assay at Cosmo Genetech (Sungsoo, Seoul, Korea). Briefly, cell protein extracts were prepared using radioimmunoprecipitation assay buffer (Thermo Scientific, Pierce, Rockford, IL, USA). Twenty micrograms of protein lysate were separated by electrophoresis on a 10% sodium dodecyl-sulfate (SDS)-polyacrylamide gel electrophoresis (PAGE) gel (Bio-Rad, Hercules, CA, USA) and transferred to a nitrocellulose membrane (Hybond-C Extra; GE Healthcare, Piscataway, NJ, USA). Nonspecific binding was blocked with 5% milk and incubated with 1:1000 primary antibody (LDLR Rabbit-mAb, #ab52818, Abcam) overnight at 4 °C. The secondary antibodies (horseradish peroxidase [HRP]-conjugated mouse anti-rabbit IgG, #SC-2357, Santa Cruz) were diluted 1:3000 in a fresh blocking solution. Protein bands were detected using the SuperSignal West Femto Kit (Thermo Fisher Scientific Inc., Waltham, MA, USA).

### 5.10. Animal Experiments

Animal studies were carried out following the guidelines for animal care approved by the Institutional Animal Care and Use Committee at the Korea Research Institute of Chemical Technology (approval number: 2020-6A-10-01). Nine-week-old male C57BL/6 mice (Orient Bio, Sungnam, Korea) were randomly divided into three groups: a control group that was fed MCS diet, the NASH model group, and the treatment group that was fed an MCD diet with once-daily oral administration of vehicle (0.5% carboxymethyl cellulose) or 30 mg/kg KRX-104130. After 4 weeks of feeding, the mice were fasted overnight, and blood and tissue samples were obtained for biochemical and histological analyses. Liver TG levels and serum concentrations of liver injury markers, including ALT and AST, were determined using a colorimetric kit (MAK266, Sigma-Aldrich, St. Louis, MO, USA) or biochemical analyzer (AU480, Beckman Coulter, Atlanta, GA, USA), respectively, according to the manufacturer’s instructions. The left lobe of the liver was fixed in 10% neutralized buffered formalin and embedded in paraffin. Histological examination was performed using hematoxylin and eosin (H&E) and picrosirius red staining of 5 µm-thick sections.

## Figures and Tables

**Figure 1 ijms-23-03807-f001:**
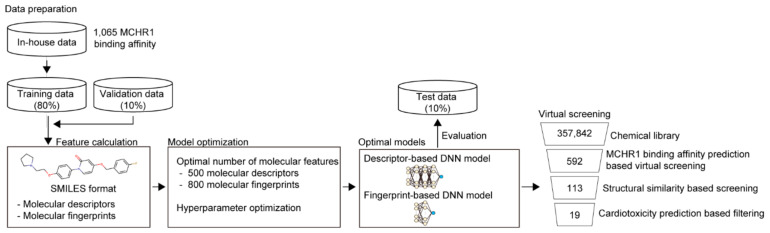
Overall scheme of virtual screening of MCHR1 antagonists.

**Figure 2 ijms-23-03807-f002:**
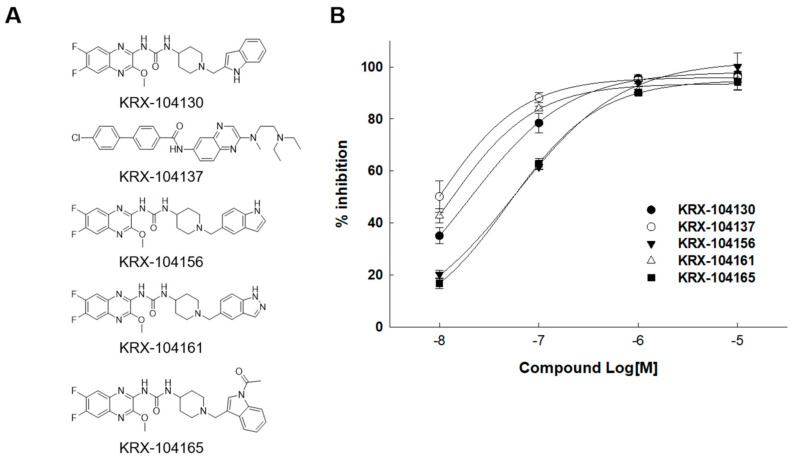
Results of MCH activity of compounds. (**A**) Structures of five compounds with MCH activity and (**B**) their inhibition rate.

**Figure 3 ijms-23-03807-f003:**
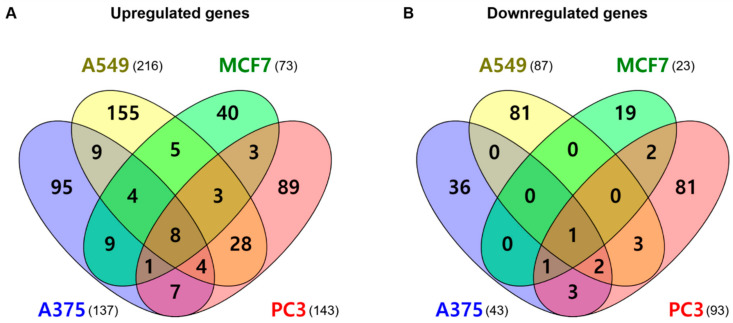
Venn diagram for the differentially expressed gene (DEG) list in four cell lines. (**A**) Upregulated genes and (**B**) downregulated genes. The crossing areas show the commonly changed DEGs. Venny website was used for drawing Venn diagrams (https://bioinfogp.cnb.csic.es/tools/venny/index.html, accessed on 1 October 2021).

**Figure 4 ijms-23-03807-f004:**
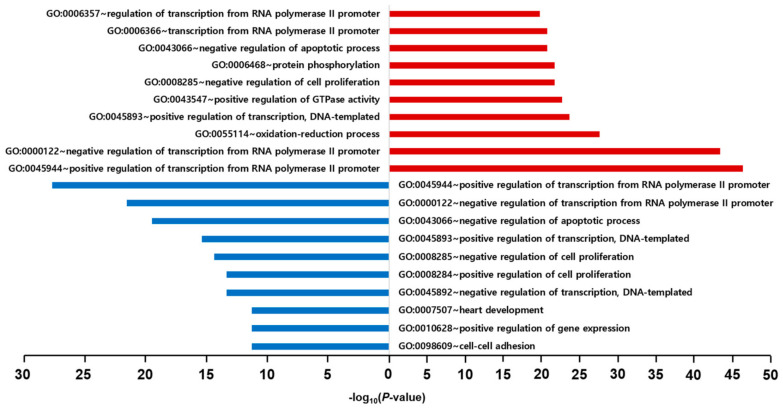
Results of GO enrichment analysis for Biological Process. The red and blue bars represent the results of GO enrichment analysis using upregulated genes and downregulated genes, respectively.

**Figure 5 ijms-23-03807-f005:**
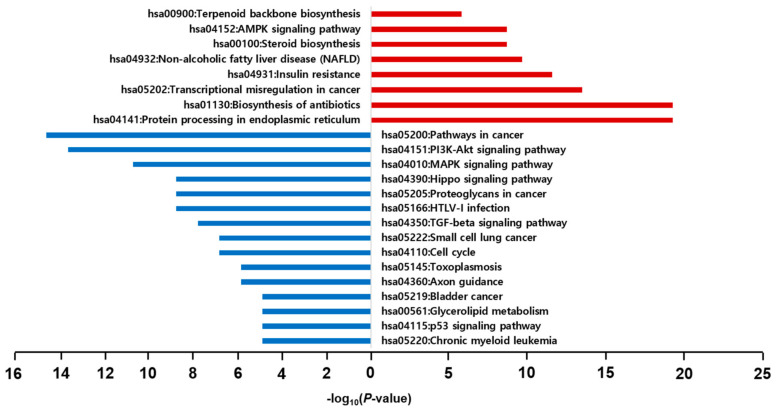
Results of KEGG pathway enrichment analysis. The red and blue bars represent the results of KEGG pathway analysis using upregulated genes and downregulated genes, respectively.

**Figure 6 ijms-23-03807-f006:**
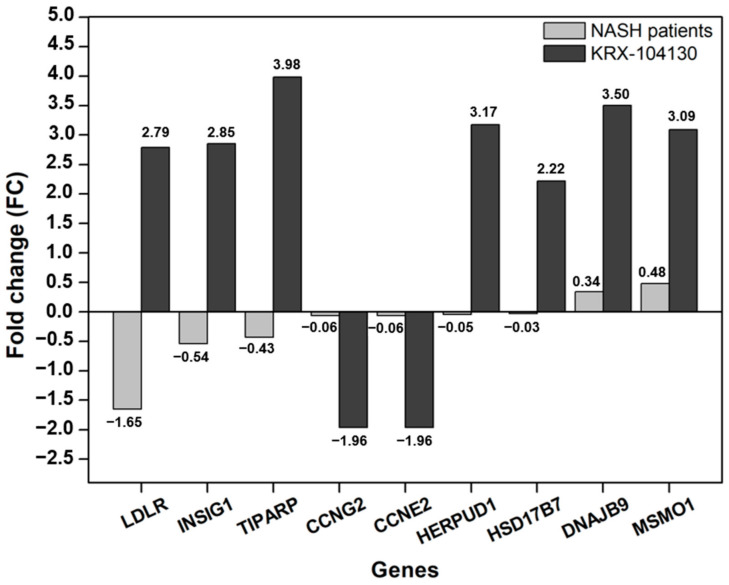
Comparison gene expression of selected common genes with NASH patient group. The *X*-axis represents the common genes for cell lines, the *Y*-axis represents the fold change. The gray and dark gray bars represent the fold change in the NASH patient group and the KRX-104130, respectively. From the left, the genes were sorted in ascending order by the fold change of the NASH patients group.

**Figure 7 ijms-23-03807-f007:**
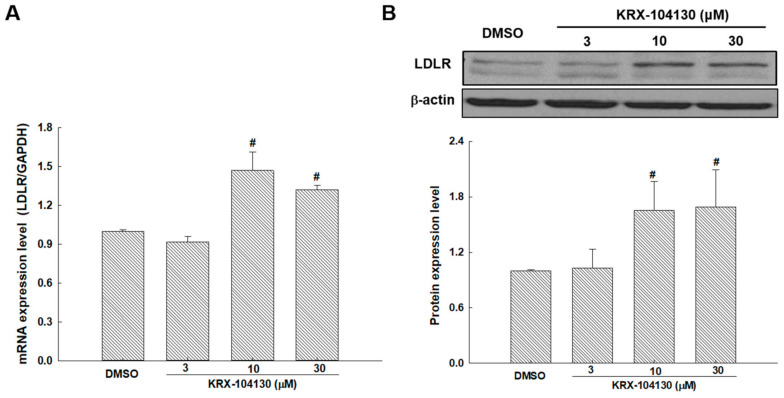
(**A**) mRNA and (**B**) protein expression levels of *LDLR* treated with 3, 10, and 30 μM KRX-104130 in the HepG2 cells. ^#^
*p* < 0.05, relative to control (*n* = 3).

**Figure 8 ijms-23-03807-f008:**
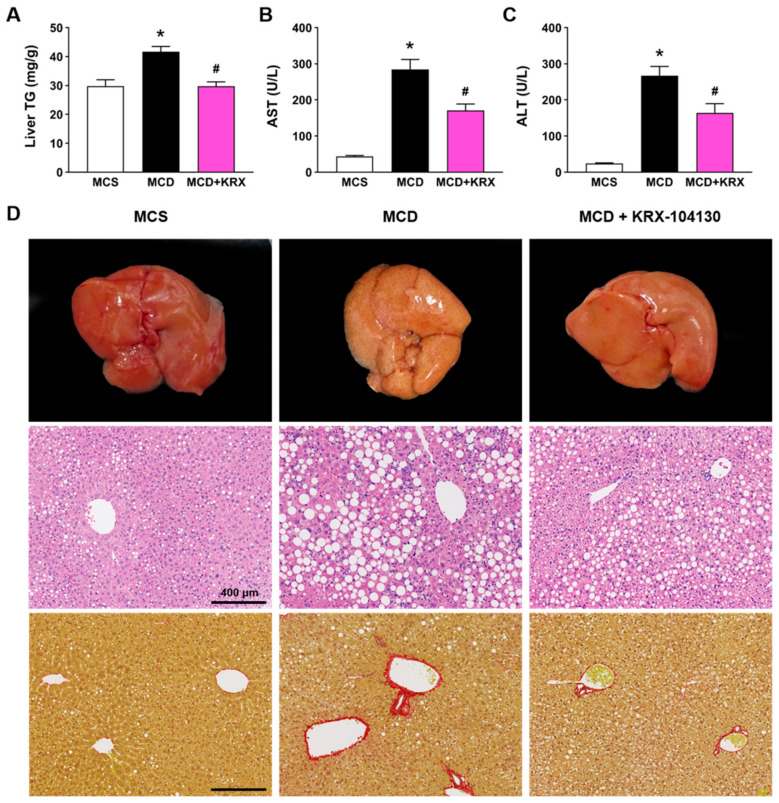
Effect of KRX-104130 treatment in a mouse model of NASH. The MCD diet-induced hepatic injury, steatosis, and fibrosis were ameliorated by treatment with KRX-104130. (**A**) Liver TG levels; (**B**,**C**) Serum ALT and AST levels; (**D**) Representative gross liver appearance (upper panel), and liver sections stained with hematoxylin and eosin (middle panel) and picrosirius Red (bottom panel) in different groups of mice. Data are presented as mean ± S.E.M. Statistical differences between experimental groups were determined by One-way ANOVA (*n* = 6–10 for each group). * *p* < 0.05 vs. MCS group; ^#^
*p* < 0.05 vs. MCD group. TG, triglycerides; AST, aspartate aminotransferase; ALT, alanine aminotransferase; MCD, methionine-and choline-deficient; MCS, methionine-choline sufficient.

**Table 1 ijms-23-03807-t001:** In vivo mouse experimental results of KRX-104130.

Compounds	KRX-104130	
Route	I.V.	P.O.
Dose (mg/kg)	10	20
n	3	3
AUC_0–t_ (μg∙h/mL)	4.53 ± 0.22	1.72 ± 0.85
T_1/2_ (h)	5.71 ± 1.03	10.22
T_max_ (h)		6.67 ± 2.31
C_max_ (μg/mL)		0.13 ± 0.05
CL (L/kg/h)	2.21 ± 0.11	
Vss (L/kg)	14.16 ± 0.72	
F (%)		20.3

## Data Availability

The datasets generated and analyzed during the current study are included in the present manuscript and Appendix A.

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
