# Peer review of "Identification and New Indication of Melanin-Concentrating Hormone Receptor 1 (MCHR1) Antagonist Derived from Machine Learning and Transcriptome-Based Drug Repositioning Approaches"

_ijms, 2022, doi:10.3390/ijms23073807_

Round 1
Reviewer 1 Report
Please refer to the attached document.

Reviewer 2 Report
In this article the authors identified a novel MCHR1 antagonist generated through virtual screening, named as KRX-104130, that reduced hepatic lipids and liver injury in a NASH mouse model without cardiotoxicity effect. At molecular level KRX-104130 antagonist increased the expression of LDLR in HepG2 cells in a concentration-dependent manner.
The work is clearly presented, and the authors provide a clear rationale of the study, however, the manuscript has some mistakes and the discussion section in my opinion should be improved before publication in this journal.
Major revision:
- In the experiment where the authors administered KRX-104130 in a NASH mouse model, could be of interest to show other parameters such food intake and body weigh during the treatment with KRX-104130 in these mice, because it might help to know a little more if these benefits observed in the liver could be extended to other tissues. If these data are available, it could be included in the study and discussed in the work. Speculating about it in the discussion could also contribute significantly to the study.
- LDLR marker could be considered the main target on which KRX-104130 acts through its effects on MCHR1 in the different in vitro or in silico studies carried out in this study, however this marker or other markers related to lipolysis are not corroborated in the NASH model. It would be very interesting to be able to obtain this data or speculate more about it in the discussion to clarify the mechanism of action of KRX-104130.
Minor comments:
- In the section 2.4, in the pharmacokinetic study, is not clear if the antagonists were administered to mice or rat, because both models appear in the description of this section even in the Discussion is confusing. Please check it.
- It is unusual to show the Material & Methods section between discussion and conclusion. I suggest to organize the conclusion section following the discussion.
- The legend of figure 8 must be written in more detail for the correct understanding of the figure.
